# NCDB Analysis of Melanoma 2004–2015: Epidemiology and Outcomes by Subtype, Sociodemographic Factors Impacting Clinical Presentation, and Real-World Survival Benefit of Immunotherapy Approval

**DOI:** 10.3390/cancers13061455

**Published:** 2021-03-22

**Authors:** Sunny R. K. Singh, Sindhu J. Malapati, Rohit Kumar, Christopher Willner, Ding Wang

**Affiliations:** 1Henry Ford Cancer Institute, Henry Ford Health System, Detroit, MI 48202, USA; cwillne1@hfhs.org (C.W.); dwang1@hfhs.org (D.W.); 2Department of Hematology and Oncology, Ascension St John Hospital, Detroit, MI 48236, USA; sindhu.malapati@ascension.org; 3Department of Hematology and Oncology, University of Louisville, Louisville, KY 40202, USA; drrk22@gmail.com

**Keywords:** melanoma, epidemiology, healthcare disparities, immunotherapy, immune checkpoint inhibitor, real-world data

## Abstract

**Simple Summary:**

Melanoma is clinicopathologically a heterogeneous disease with rising incidence. Metastatic disease is associated with poor outcomes, and immunotherapy was first approved in 2011 for its treatment. In our analysis of a large national database, we describe the epidemiology, clinical presentation, and survival outcomes of cutaneous, ocular, and mucosal melanoma in recent years. Metastatic cutaneous melanoma had better survival than both metastatic ocular and mucosal melanoma. We found higher odds of metastatic disease at diagnosis amongst African Americans compared to Caucasians. Additionally, for metastatic cases, we noted 25% lower mortality in those treated at an academic facility compared to community cancer programs and a 20% real-world survival benefit following approval of immunotherapy. This real-world survival benefit was definitely seen in Caucasians and those with cutaneous or mucosal melanoma. Further investigation is needed to confirm this benefit in African Americans and ocular melanoma.

**Abstract:**

**Background:** The incidence of invasive melanoma is rising, and approval for the first immune checkpoint inhibitor (ICI) to treat metastatic melanoma occurred in 2011. We aim to describe the epidemiology and outcomes in recent years, sociodemographic factors associated with the presence of metastasis at diagnosis, and the real-world impact of ICI approval on survival based on melanoma subtype and race. **Methods:** This is a retrospective analysis of the National Cancer Database (NCDB) from the years 2004–2015. The primary outcome was the overall survival of metastatic melanoma by subtype. Secondary outcomes included sociodemographic factors associated with the presence of metastasis at diagnosis and the impact of treatment facility type and ICI approval on the survival of metastatic melanoma. **Results:** Of the 419,773 invasive melanoma cases, 93.80% were cutaneous, and 4.92% were metastatic at presentation. The odds of presenting with metastatic disease were higher in African Americans (AA) compared to Caucasians (OR 2.37; 95% CI 2.11–2.66, *p* < 0.001). Treatment of metastatic melanoma at an academic/research facility was associated with lower mortality versus community cancer programs (OR 0.75, 95 % CI 0.69–0.81, *p*-value < 0.001). Improvement in survival of metastatic melanoma was noted for Caucasians after the introduction of ICI (adjusted HR 0.80, 95% CI 0.78–0.83, *p* < 0.001); however, this was not statistically significant for AA (adjusted HR 0.80, 95% CI 0.62–1.02, *p*-value = 0.073) or ocular cases (HR 1.03, 95% CI 0.81–1.31, *p*-value = 0.797). **Conclusion:** Real-world data suggest a 20% improvement in survival of metastatic melanoma since the introduction of ICI. The disproportionately high odds of metastatic disease at presentation in AA patients with melanoma suggest the need for a better understanding of the disease and improvement in care delivery.

## 1. Introduction

Melanoma is the fifth most common cancer in the US [1]. A recent Surveillance, Epidemiology, and End Results (SEER) combined database analysis from 2001 to 2015 demonstrated a steady increase in cases of invasive melanoma between 2001 and 2015 [2]. Melanoma is broadly classified into two categories, cutaneous and noncutaneous. While noncutaneous melanoma has a lower incidence, it is characterized by a much higher proportion of patients presenting with invasive or metastatic disease compared to cutaneous melanoma. Of 82,943 cases of melanoma with a known primary site, reported to the National Cancer Data Base (NCDB) in the United States for 1985–1994, only 5.5% were ocular, and 1.3% occurred at mucosal sites [3]. According to a report based on data from the North American Association of Central Cancer Registries, only 6691 cases of noncutaneous melanoma (4885 ocular and 1806 mucosal) were diagnosed among 851 million person-years at risk between 1996 and 2000 [4]. The rarity of noncutaneous melanoma is the most likely explanation for the paucity of large descriptive studies in this area. We, therefore, leveraged the NCDB, which contains 70% of all newly diagnosed cancers in the US, and included cases diagnosed within the two last decades to perform such an analysis. Melanoma is denoted by immunogenic biology and has, thus, been the focus of research, clinical trial developments, and breakthroughs utilizing immune checkpoint inhibitors (ICIs). This led to Food and Drug Administration (FDA) approval of ipilimumab (Yervoy), an anti-cytotoxic T lymphocyte associated protein-4 (anti-CTLA4) antibody, in 2011 for the treatment of newly diagnosed or previously treated unresectable/metastatic melanoma. This was followed by the approval of programmed cell death protein-1 (PD-1) inhibitors, nivolumab (Opdivo) and pembrolizumab (Keytruda), in 2014 for the treatment of unresectable or metastatic melanoma. Since these FDA approvals, the National Comprehensive Cancer Network (NCCN) has recommended the use of ICIs for unresectable/metastatic melanoma treatment, solidifying them as a standard of care. This includes the treatment of metastatic ocular and mucosal melanomas, done in accordance with guidelines for cutaneous metastatic melanoma [5]. There is a paucity of survival data concerning noncutaneous metastatic melanoma, both before and after the introduction of these novel agents.

In our study, we carry out an in-depth descriptive analysis of the epidemiology, prevalence, and outcomes of cutaneous, ocular, and mucosal melanoma in the United States, using a large cohort of patients from the NCDB, with data collected from 2004 to 2015. We evaluate the association between socioeconomic–demographic factors and the presence of metastatic disease at diagnosis. We also investigate the impact of treatment facility type on mortality of metastatic melanoma and compare survival of patients diagnosed with metastatic melanoma and its subtypes before and after the introduction of ipilimumab in 2011, the first ICI approved for this indication. In addition, using real-world data, we attempt to assess the impact of socioeconomic factors, including levels of education and household income, type of insurance coverage, and race, on the survival benefit following the approval of ICIs for treatment of metastatic melanoma.

## 2. Methods

### 2.1. Data Source

This is a retrospective cohort study using hospital registry data collected between 2004 and 2015 within the National Cancer Database (NCDB). The NCDB is jointly sponsored by the Commission on Cancer of the American College of Surgeons and the American Cancer Society and includes cancer cases from more than 1500 commission-accredited cancer programs [6]. This study was determined to be exempt from full review by the Henry Ford Health System Institutional Review Board (IRB number: 13,331).

### 2.2. Study Population

The study selection criteria are outlined in Figure 1. We included patients ≥18 years of age who had the appropriate morphology and disease stage using International Classification of Diseases for Oncology 3rd Edition (ICD-O-3) codes for invasive melanoma. Based on ICD-O-3 topographical codes of primary sites for skin and noncutaneous sites, we divided the study population into two groups, cutaneous and noncutaneous melanoma [7]. Only patients with a known primary site were included in this study. The noncutaneous group was further divided into two major subtypes, ocular and mucosal. The cohort of mucosal melanoma patients included those with invasive melanoma arising in the head and neck, anorectum, gastrointestinal tract (excluding anorectum) and genitourinary system. These categories were generated using conventionally used categorization schema in the literature [3,8,9,10,11]. All the ICD-O-3 codes used for the purpose of this study and their descriptions are provided in Appendix A. The NCDB Participant User File (PUF) dictionary was used to define variables for the study [12]. American Joint Committee on Cancer (AJCC) staging was applied to melanomas of the skin only, given the absence of a standardized staging system for most of the noncutaneous melanoma subtypes. We, therefore, collected data regarding lymph node status (positive/negative/unknown status or not examined) and the presence or absence of distant metastasis at the time of diagnosis to define the extent of disease. The main histologic groupings, representing cutaneous melanomas, were nodular, lentigo maligna, superficial spreading, acral lentiginous, and melanoma, not otherwise specified (NOS). The remaining histologies were categorized as “other”. In addition to tumor site, histology, and extent of disease, data regarding patient characteristics, including age, gender, race, and chronic comorbid conditions quantified by Charlson/Deyo score, were collected. Socioeconomic demographics, including insurance status, median family income by zip code, proportion of residents without a high school diploma by zip code, and geographical region of United States, were collected. Treatment and care accessibility-related factors, including structural characteristics of the facility delivering the care (i.e., community cancer program, academic/research cancer program, or integrated network cancer program) and treatment modality received (systemic therapy, radiotherapy, or surgery), were also collected.

### 2.3. Outcomes and Statistical Analysis

We conducted a descriptive analysis of the baseline clinicopathologic features of the included patients and used the chi-square test to compare the characteristics of patients by subgroup.

### 2.4. Primary Outcome

Overall survival (OS) of metastatic melanoma by subtype: OS was defined as the number of months from the patient’s date of diagnosis to either their date of death or when they were lost to follow-up, as the NCDB does not collect cancer-specific survival. The cumulative OS rates and median OS were estimated using Kaplan–Meier analysis for each of the cutaneous, ocular, and mucosal melanoma subgroups.

### 2.5. Secondary Outcomes

Association of socioeconomic and demographic factors with the presence of metastatic disease at diagnosis: Multivariate logistic regression was used to analyze factors including age, sex, race, comorbidity burden, degree of education, income level, insured status, geographical location, and treatment facility type.

Impact of treatment facility type on survival in metastatic melanoma: This was analyzed using multivariate logistic regression to distinguish potential differences between the outcome of treatment at institutions identified as academic, community, and integrated network cancer programs.

Impact of the introduction of ICIs on survival in metastatic melanoma: Using the year of approval of ipilimumab as a landmark (2011), we compared the survival of metastatic melanoma cases diagnosed between 2004 and 2010 against those diagnosed between 2011 and 2015. We also analyzed the impact of race on this outcome.

All *p*-values were two-sided. *p*-values less than 0.05 were considered statistically significant. Statistical analysis was performed using Stata/IC version 15.1 (StataCorp LP, College Station, TX, USA).

## 3. Results

### 3.1. Baseline Characteristics

Of the 11,970,999 adult cancer patients in the database, 552,989 (4.62%) had melanoma, of which 419,773 (75.91%) had invasive melanoma. Only patients with invasive disease were included in this study. As shown in Figure 1, cases of invasive melanoma could be subdivided into cutaneous melanoma (93.80%) and noncutaneous melanoma (6.20%). The noncutaneous cohort was further divided into ocular (68.06%) and mucosal (28.57%) subtypes. Table 1 shows the baseline characteristics of these cohorts. The main histologic groupings of cutaneous melanoma included superficial spreading (29.61%), nodular (9.31%), lentigo maligna (5.10%), and acral lentiginous (1.31%). The remaining groups were melanoma, NOS (50.31%), and other (4.36%). The majority cases (67.38%) of cutaneous melanoma were Stage 1 at diagnosis (Appendix A). Table 2 shows the mortality and extent of disease based on the histological subtypes of cutaneous melanoma, while Table 3 analyzes the differences in key characteristics of cutaneous melanoma based on race. Amongst cases of ocular melanoma, 90.28% (*n* = 15,992) originated in the uvea, while 4.95% (*n* = 877) originated in the conjunctiva. Amongst cases of mucosal melanoma, 43.78% (*n* = 3255) originated in the genitourinary (GU) tract, 34.82% (*n* = 2589) in the head and neck (H&N) region, 18.17% (*n* = 1351) in the anorectal part of the gastrointestinal tract (GIT) and 3.23% (*n* = 240) in the GIT, excluding anorectum. Table 4 compares the baseline characteristics of the mucosal subtypes. The most common subtypes of H&N melanoma included the nasal/paranasal (72.85%, *n* = 1886) and oral (21.55%, *n* = 558). Amongst the GU melanoma cases, the most common type included female GU (90.48%, *n* = 2945), followed by male GU (5.04%, *n* = 164) and urinary tract (4.49%, *n* = 146) melanoma.

Disease burden between years 2004 and 2015 for invasive melanoma subtypes: As shown in Figure 2 and Appendix A, the total number of cases increased over the study period in all three subtypes of melanoma. Trends in the proportion of patients diagnosed with metastasis at diagnosis did not appear to undergo significant changes.

Metastatic melanoma: Of all invasive melanoma cases, 20,691 (4.92%) were found to have metastasis at presentation. Of these, the majority were of cutaneous origin (94.20%, *n* = 19,492), followed by mucosal (4.22%, *n* = 873) and ocular (1.57%, *n* = 326). The mean age of patients with metastasis at diagnosis was 64.36 years for cutaneous cases versus 67.29 years for noncutaneous cases (*t*-test *p*-value < 0.001). The proportion of African American (AA) patients amongst noncutaneous cases of metastatic melanoma was higher (6.34%, *n* = 76) when compared to cutaneous (1.71%, *n* = 333) cases (*p*-value < 0.001). Overall mortality was high amongst all subtypes (Appendix A). It was highest for mucosal cases (87.51%), followed by ocular (85.28%) and cutaneous (80.04%) cases (*p*-value < 0.001).

### 3.2. Primary Outcome

Survival in patients with metastatic melanoma: Among those diagnosed with metastasis at presentation, median OS was 8.97 months for the cutaneous group, 9.10 months for the ocular group, and 8.38 months for the mucosal group. As shown in Table 5 and Figure 3A,B, the cutaneous group had better survival than both ocular and mucosal groups, while no difference was noted between the survival of ocular vs. mucosal melanoma. To give perspective, when we analyzed median OS for all cases, it was 161.12 months (about 13.5 years) in cutaneous melanoma, 122.91 months (about 10 years) in ocular melanoma, and 26.29 months (about 2 years) in mucosal melanoma. Furthermore, when we specifically analyzed nonmetastatic cases of cutaneous melanoma by stage, median OS was not reached for Stage 1; it was 86.41 months (about 7 years) for Stage 2 and 61.37 months (about 5 years) for Stage 3.

### 3.3. Secondary Outcomes

Impact of sociodemographic factors on odds of metastatic disease at diagnosis: Factors analyzed included patient factors (age, race, sex, burden of comorbidities), melanoma type, sociodemographic factors (education, income, insurance, geographical location), and hospital type. We first analyzed the above factors using a univariate logistic regression model. A multivariate regression model was then developed by including the factors, which yielded a *p*-value < 0.2 on univariate analysis (Table 6). Factors associated with higher odds of presenting with metastatic disease at diagnosis included increasing age, male sex, and AA race. While socioeconomic status had no association with this outcome, a rising level of education correlated with progressively lower odds of presenting with metastatic disease. Interestingly, patients being treated at community cancer programs had higher odds of presenting with metastatic disease when compared to academic/research programs.

Impact of treatment facility type on mortality in patients with metastatic melanoma: We analyzed the impact of treatment facility type on mortality amongst all cases of metastatic melanoma using multivariate logistic regression analysis after adjusting for melanoma type and other sociodemographic factors (Appendix A). We noted that treatment at an academic program was associated with lower mortality (OR 0.75, 95% CI 0.69–0.81, *p*-value < 0.001) in contrast to a community cancer program. Appendix A analyzes the differences amongst the facilities in terms of the patients’ race, income, and level of education.

Overall survival after the approval of ICI for metastatic melanoma: We compared the survival of patients who were metastatic at diagnosis, before and after the approval of immunotherapy (2011 onwards) using the Cox proportional hazard model. The adjusted hazards for survival in patients with metastatic melanoma after the introduction of ipilimumab (years 2011–2015) was 0.81 (95% CI 0.78–0.83, *p*-value < 0.001) when compared to survival prior to its approval (years 2004–2010) (Appendix A). Table 5 shows the median OS and 3-year survival of metastatic cases for years 2004–2010 and 2011–2015. As shown in the table, median OS during the years 2011–2015 versus 2004–2010 was significantly improved in the cutaneous and mucosal groups. Kaplan–Meier survival curves for survival analysis are shown in Figure 3C,D.

Impact of race upon survival in melanoma following ICI approval: To study the impact of race (in Caucasians and African Americans) on the survival benefit since the approval of immunotherapy, we used multivariable Cox regression for the respective cohorts. We adjusted for patient factors (age, race, sex, burden of comorbidities), melanoma type, sociodemographic factors (education, income, insurance, geographical location), and facility type (Appendix A). We noted a 20% improvement in survival for Caucasians with the introduction of immunotherapy (adjusted HR 0.80, 95% CI 0.77–0.83, *p* < 0.001); however, the difference did not reach statistical significance for African American patients (adjusted HR 0.80, 95% CI 0.62–1.02, *p*-value = 0.073).

## 4. Discussion

In this evaluation of real-world data, we present the results of a retrospective analysis of the largest cohort of invasive melanoma cases (*n* = 419,773) diagnosed between 2004 and 2015. Cutaneous melanoma accounted for 93.80% of cases, followed by ocular (4.22%) and then mucosal (2.00%) melanomas. The most common anatomic site of involvement in ocular melanoma was the conjunctiva (90.28%), and in mucosal melanoma, the genitourinary tract (43.78%). During this 10-year period of NCDB data collection, the number of cases increased by 61.04% for cutaneous melanoma, by 39.27% for ocular melanoma, and by 59.07% for mucosal melanoma. These findings are similar to those of other investigations, one of which additionally concluded a doubling in melanoma incidence from 1982 to 2011 [13]. Notably, the proportion of those with metastatic disease at diagnosis did not change drastically over the period of study. The nature of this dataset prevents us from making definitive conclusions concerning the increasing incidence of melanoma. However, at least from the standpoint of cutaneous melanoma, we can infer that majority of the patients come from a population or generation that experienced a high degree of lifetime sun-exposure and damage. This was prior to the widespread public adoption of skin cancer and melanoma prevention strategies, particularly using FDA-regulated sunscreen protection products, which have been more often utilized in accordance with recommendations from the American Academy of Dermatology (AAD).

There were notable differences among melanoma subtypes in terms of demographic and socioeconomic factors. In our analysis, we found that both cutaneous and ocular melanoma were frequently diagnosed at younger ages, in the subgroup aged 31–64 years (51.33% and 51.67%, respectively), while mucosal melanoma patients were primarily aged 65 years or older at diagnosis (62.03%). Patients with mucosal melanoma were significantly more likely to be female (70.44% female), while in both cutaneous and ocular melanoma, there was a slight male predominance. The higher proportion of females in the mucosal melanoma subgroup is attributable to females accounting for 93.89% of the most common subtype of mucosal melanoma, genitourinary melanoma. Of these, 90.48% were of vulvar or vaginal origin. More than half of the patients in the cutaneous melanoma subgroup (52.53%) were privately insured, while 58.72% of patients in the mucosal subgroup had Medicare or other government insurance. This may be attributable to more patients being aged 65 or older in the mucosal melanoma subgroup. Patients diagnosed with cutaneous melanoma had a higher level of education and resided in zip codes with a higher median household income. It has been hypothesized that those with less secondary education perceive themselves to be less likely to develop skin cancer [14]. Caucasians constituted an overwhelming majority of patients in all three groups (89.86–97.39%). African Americans made up a significantly smaller proportion of patients in the overall cohort, most notably comprising 0.6% of cutaneous melanoma cases. However, even with lower incidence, their proportion was higher in noncutaneous melanoma, comprising 5.11% of mucosal melanomas.

There was considerable variation in the extent of disease (positive lymph nodes or metastasis) between the groups at diagnosis, stratified by anatomic site. Positive lymph nodes were identified at presentation in 11.64% of cutaneous and 16.76% of mucosal melanoma patients but in only 0.11% of ocular melanoma cases. While sentinel lymph node assessment is not standard of care in ocular melanoma, the low frequency of positive lymph nodes in the ocular melanoma cohort could also be explained by the absence of lymphatic supply in the uvea [15]. Metastatic disease was identified in 5.11% of cutaneous and 1.90% of ocular melanoma patients and in 12.61% of mucosal melanoma cases. Notable factors associated with lower odds of metastatic disease at presentation included female gender, higher level of education, and lower burden of comorbidity. Interestingly, socioeconomic factors did not seem to impact this outcome. Additionally, compared to community cancer programs, academic/research cancer centers were 24% less likely to encounter cases that were metastatic at diagnosis. This observation could possibly be explained by referral bias and a higher index of suspicion coupled with better logistics in terms of diagnostic capabilities and longitudinal follow-up at academic/research institutions.

The low rate of metastatic disease in ocular melanoma may be attributable to routine ocular examinations, leading to increased detection rates. About 30–40% of ocular melanomas are asymptomatic and detected on a routine ocular examination [15]. Hidden location, late onset of symptoms, potential amelanotic appearance, the presence of a rich lymphovascular supply are some of the suspected causes that contribute to a delay in diagnosis and more advanced disease at presentation of mucosal melanoma [16,17]. Within the mucosal melanoma group, the subtype with the highest proportion of metastatic disease and positive lymph nodes at diagnosis included GIT (excluding anorectum) melanoma, in which about one-third (33.03%) were metastatic and around one-fourth (24.35%) had positive lymph nodes at diagnosis.

The odds of presenting with metastatic disease were higher in African Americans compared to Caucasians (OR 2.37; 95% CI 2.11–2.66, *p* < 0.001) after controlling for melanoma type and sociodemographic factors. Metastatic disease was noted at presentation in 14.86% of African American patients but in only 5.03% of Caucasians (chi-squared *p*-value < 0.001). Our findings are similar to published literature, substantiating significantly higher odds of African Americans having a more advanced stage at diagnosis [14]. Specifically, from the perspective of cutaneous melanoma, ulceration was present in 41.86% of AA patients versus 18.28% Caucasians. Additionally, a Breslow depth of ≥4 mm was noted in 26.28% of AA patients versus 9.25% Caucasians (Table 3). Some authors postulate (based on surveyed African American patients) that this may relate to decreased awareness of susceptibility to skin cancers in this population [14]. Moreover, there are differences in disease biology. For example, acral lentiginous melanoma (ALM), which is typically associated with a more aggressive course, is found in a greater proportion amongst African Americans compared to Caucasians [18]. This was also observed in our analysis of cutaneous melanoma, where 19.52% of African Americans (while only 1.15% of Caucasians) had a histological diagnosis of ALM (Table 3). The poorer prognosis of ALM is thought to be multifactorial, including differences in embryonic origin and immune microenvironment of glabrous versus nonglabrous skin, differing mutational landscape (age-associated signatures, dominated by chromosomal instability, instead of the more traditional UV-induced signatures), and the difficulty of making an early diagnosis of ALM [19,20,21].

As expected, the therapeutic modalities utilized also varied based on the melanoma subtype. The major therapeutic modality employed in patients with cutaneous and mucosal melanoma was surgery (93.58% and 83.72%, respectively). Radiation therapy was most often utilized in patients with ocular melanoma (69.33%). The frequency with which each modality was utilized in our analysis is consistent with current practice guidelines [22,23]. The mucosal subgroup had the highest proportion of patients (22.17%) receiving systemic therapy as a part of treatment (versus 3.47% and 7.53% in ocular and cutaneous melanoma, respectively). This may be due to a higher proportion of metastatic cases at presentation amongst mucosal melanoma. Within the mucosal cohort, the anorectal subtype had the highest proportion of patients undergoing systemic therapy (34.51%). In addition to the variation in type of treatment administered, the type of facility where the patients received care also differed. While 41.61% of the cutaneous melanoma cases were treated in community centers, the majority of cases of ocular (77.49%) and mucosal melanoma (51.94%) received treatment at an academic/research institution, and we postulate this reflects the unique, highly specialized care needs of these patients. Treatment at such centers may have provided these patients with access to other immune checkpoint inhibitor-based treatments through clinical trial enrollment. The abovementioned findings could, in part, be the reason for a higher proportion of AA patients receiving care in academic centers versus community centers (Appendix A)

Mortality greatly varied based on melanoma type and presence or absence of metastasis. The presence of metastasis was associated with a greater than 80% mortality across all groups of melanomas (Appendix A). Amongst patients presenting with metastatic disease at onset, the risk of death for those with metastatic mucosal melanoma was higher by 17% when compared to metastatic cutaneous melanoma. Similarly, the risk of death for metastatic ocular melanoma was higher by 14% when compared to metastatic cutaneous melanoma. This translated to a 3-year OS rate of 20.32% in the metastatic cutaneous melanoma group versus 13.26% in metastatic ocular and 10.52% in metastatic mucosal groups. There was no difference in survival of metastatic cases of mucosal versus ocular melanoma. Using multivariate analysis, we examined the association of treatment facility type on mortality of patients with metastatic melanoma and found that treatment at an academic facility was associated with 25% lower mortality than in community cancer programs. This could possibly be explained by the lower odds of presenting with metastatic disease (as was noted above), the availability of novel therapies within clinical trials, and robust care logistics at academic/research institutions. Our investigation revealed a 20% decrease in mortality for metastatic melanoma in the real-world setting following the approval of immunotherapy (ICIs) in 2011. These findings are similar to clinical efficacies reflected in the long-term survival follow-ups reported from the initial Phase 2 and Phase 3 clinical trials investigating ipilimumab. For example, ipilimumab was shown to produce a survival of ≥2 years in 20% of pretreated patients [24]. We did not find a statistically significant improvement in survival following the introduction of ICIs amongst AA patients; this may be due to the small sample size of the AA patient cohort. When analyzed by melanoma subtypes, the clinical benefit of ICI therapy was not noted within the metastatic ocular melanoma subtype as well (HR 1.03, 95% CI 0.81–1.31, *p*-value = 0.797). Patients with metastatic cutaneous and mucosal melanoma noted an improved survival of 18% and 26%, and this translated to a median OS prolongation of 2.24 and 3.35 months, respectively. We postulate these differences could be due to variations in disease biology of the melanoma subtypes.

There are several limitations of this study. These include its retrospective design, the paucity of data in terms of the specifics of systemic therapy used, and molecular/genetic characteristics. Though ipilimumab was the first ICI to be FDA approved in 2011 as standard of care ICI for metastatic melanoma, a small cohort of metastatic melanoma patients might have received other ICIs (e.g., nivolumab and pembrolizumab) through clinical trial enrollment. Despite being a national database, NCDB may not be an accurate representation of the entirety of melanoma cases, as only patients from participating institutions are captured. As NCDB collects only absolute numbers of cases and is not matched to local populations, cancer incidence rates cannot be measured with this data set. Data regarding sentinel lymph node biopsy was not available for the pre-2012 years in the database and hence not reported here. Mortality reported in NCDB is from all causes rather than being cancer-specific, and this limits our ability to report disease-specific survival or relative survival. Finally, NCDB does not provide data about recurrences and progression of disease.

## 5. Conclusions

Metastatic melanoma continues to be a disease associated with high mortality. Its rising incidence underlines the urgent need for cultivating a high degree of suspicion in addition to improving our diagnostic capabilities. The disproportionately high odds of metastatic disease amongst African Americans needs to be a priority for future research and interventions, both medical and social. Our study substantiates a real-world survival benefit in patients with metastatic melanoma since the approval of immunotherapy and raises some important questions about differences in outcomes based on melanoma subtype. Our findings also underscore the importance of ensuring uniform access to novel treatments and furthering our understanding of molecular differences between melanoma subtypes and amongst patients from different racial backgrounds.

## Figures and Tables

**Figure 1 cancers-13-01455-f001:**
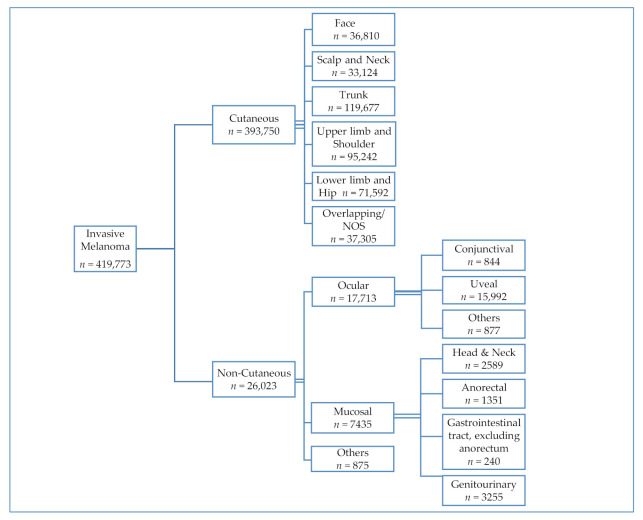
Case schema for invasive melanoma subtypes for years 2004–2015 National Cancer Data Base (NCDB).

**Figure 2 cancers-13-01455-f002:**
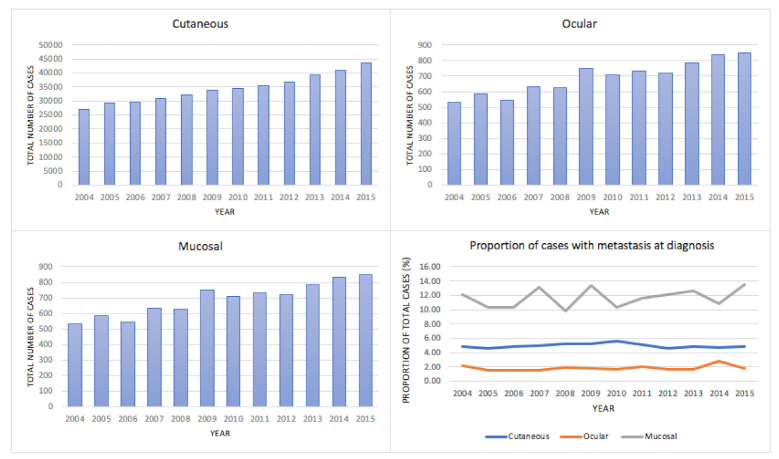
Burden of invasive melanoma between the years 2004 and 2015.

**Figure 3 cancers-13-01455-f003:**
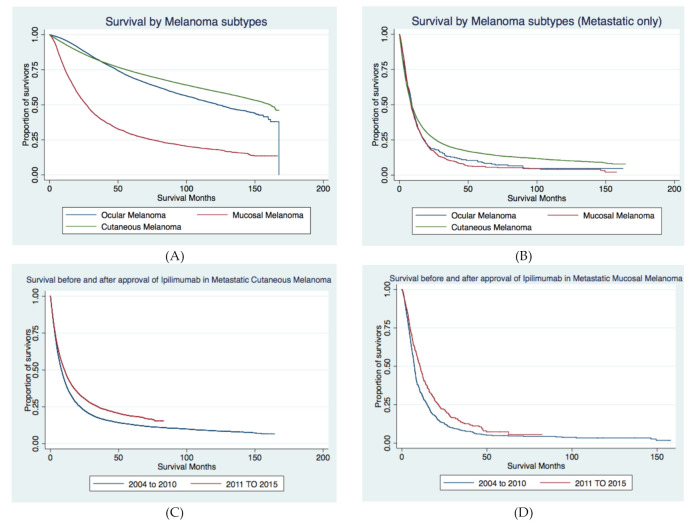
Kaplan–Meier analysis for (**A**) survival amongst all cases of invasive melanoma subtypes diagnosed between 2004 and 2015, (**B**) survival amongst melanoma cases with metastatic disease at presentation diagnosed between 2004 and 2015 by subtype, (**C**) survival pre- and postapproval of ipilimumab in 2011 amongst cases of cutaneous melanoma with metastatic disease at presentation., (**D**) survival pre- and postapproval of ipilimumab in 2011 amongst cases of mucosal melanoma with metastatic disease at presentation.

**Table 1 cancers-13-01455-t001:** Baseline characteristics of all invasive melanoma subtypes (National Cancer Database; NCDB 2004–2015).

Invasive Melanoma (All Cases)	Cutaneous Melanoma *n* = 393,750 (%)	Ocular Melanoma *n* = 17,713 (%)	Mucosal Melanoma *n* = 7435 (%)	Chi-Squared *p*-Value
**Females**	166,288 (42.23)	8491 (47.94)	5237 (70.44)	<0.001
**Age**				<0.001
≤30 Years	17,498 (4.44)	423 (2.39)	126 (1.69)	
31 to 64 Years	202,128 (51.33)	9153 (51.67)	2697 (36.27)	
≥65 Years	174,124 (44.22)	8137 (45.94)	4612 (62.03)	
**Race**				<0.001
Caucasian	383,462 (97.39)	16,861 (95.19)	6681 (89.86)	
African American	2382 (0.60)	178 (1.00)	380 (5.11)	
**Charlson Deyo score**				<0.001
0	338,934 (86.08)	14,698 (82.98)	5959 (80.15)	
1	43,252 (10.98)	2428 (13.71)	1132 (15.23)	
2	8529 (2.17)	426 (2.41)	249 (3.35)	
≥3	3035 (0.77)	161 (0.91)	95 (1.28)	
**Positive Lymph Nodes identified**	45,177 (11.64)	18 (0.11)	1211 (16.76)	<0.001
**Metastatic disease at diagnosis**	19,492 (5.11)	326 (1.90)	873 (12.61)	<0.001
**Died**	103,211 (26.21)	5388 (30.98)	4947 (66.54)	<0.001
**Insurance Status**				<0.001
Not insured	9674 (2.46)	600 (3.39)	187 (2.52)	
Private insurance	206,840 (52.53)	8486 (47.91)	2608 (35.08)	
Medicaid	10,145 (2.58)	534 (3.01)	274 (3.69)	
Medicare or other government insurance	167,091(42.44)	8093 (45.69)	4366 (58.72)	
**Education Level: *Number of adults in the patient’s zip code who did not graduate from high school***				<0.001
>29% (level 1)	27,642 (7.05)	1739 (9.86)	792 (10.70)	
20–28.9% (level 2)	69,139 (17.63)	3712 (21.06)	1557 (21.04)	
14–18.9% (level 3)	100,822 (25.71)	4836 (27.43)	1958 (26.46)	
<14% (level 4)	194,588 (49.62)	7342 (41.65)	3098 (41.80)	
**Socioeconomic Status: *Median household income for patient’s zip code***				<0.001
<40,227 (level 1)	22,270 (5.68)	1401 (7.95)	633 (8.56)	
40,227–50,353 (level 2)	55,505 (14.16)	3226 (18.30)	1224 (16.54)	
50,354–63,332 (level 3)	104,836 (26.74)	5190 (29.45)	2066 (27.92)	
>46,000 to >63,333 (level 4)	209,444 (53.42)	7808 (44.30)	3476 (46.98)	
**Treating Facility Type**				<0.001
Community cancer program	145,618 (41.61)	2361 (14.27)	2586 (36.14)	
Academic/research program	162,245 (46.36)	12,823 (77.49)	3716 (51.94)	
Integrated Network Cancer Program	42,114 (12.03)	1365 (8.25)	853 (11.92)	
**Received Radiation therapy**	16,473 (4.21)	12,256 (69.33)	2350 (31.86)	<0.001
**Underwent Surgery**	368,260 (93.58)	5852 (33.08)	6217 (83.72)	<0.001
**Received Systemic therapy**	29,415 (7.53)	612 (3.47)	1628 (22.17)	<0.001

**Table 2 cancers-13-01455-t002:** Histological subtypes of cutaneous melanoma (NCDB 2004–2015): mortality, lymph node status, and presence of metastasis at diagnosis.

		Melanoma, not Otherwise Specified *n* (%)	Acral Lentiginous *n* (%)	Nodular *n* (%)	Superficial Spreading *n* (%)	Lentigo Maligna *n* (%)	Others *n* (%)	Chi-Squared *p*-Value
**Died** **(For all cases)**	**Yes**	57,076 (28.81)	1744 (33.9)	15,246 (41.58)	18,242 (15.65)	5203 (25.89)	5700 (33.2)	<0.001
	**No**	141,022 (71.19)	3400 (66.1)	21,422 (58.42)	98,329 (84.35)	14,896 (74.11)	11,470 (66.8)	
**Lymph node status**	**Positive**	23,344 (12.08)	1223 (23.89)	9547 (26.13)	8848 (7.61)	411 (2.05)	1804 (10.63)	<0.001
	**Negative**	77,179 (39.95)	2281 (44.55)	19,007 (52.03)	47,802 (41.12)	4699 (23.49)	9359 (55.14)	
	**Unknown/Not examined**	92,655 (47.96)	1616 (31.56)	7977 (21.84)	59,596 (51.27)	14,891 (74.45)	5811 (34.23)	
**Metastasis at Diagnosis**	**Present**	16,532 (8.76)	97 (1.91)	1496 (4.18)	480 (0.42)	66 (0.33	821 (4.92)	<0.001
	**Absent**	172,266 (91.24)	4971 (98.09)	34,269 (95.82)	114,905 (99.58)	19,728 (99.67)	15,872 (95.08)	

**Table 3 cancers-13-01455-t003:** Characteristics of cutaneous melanoma based on race (NCDB 2004–2015): Ulceration, Breslow thickness, American Joint Committee on Cancer (AJCC) clinical stage, and histology subtype.

		Caucasians *n* (%)	African Americans *n* (%)	Others *n* (%)	Chi-Squared *p*-Value
**Ulceration**	**Absent**	278,767 (81.72)	1125 (58.14)	5612 (81.23)	<0.001
	**Present**	62,341 (18.28)	810 (41.86)	1297 (18.77)	
	**Total**	341,108	1935	6909	
**Breslow Thickness**	**<1 mm**	199,529 (58.49)	680 (37.55)	4265 (61.38)	<0.001
	**>1 to 2**	68,282 (20.02)	309 (17.06)	1219 (17.54)	
	**>2 to 4**	41,798 (12.25)	346 (19.11)	804 (11.57)	
	**>4**	31,540 (9.25)	476 (26.28)	661 (9.51)	
	**Total**	341,149	1811	6949	
**AJCC Clinical Stage**	**1**	173,291 (67.56)	471 (34.13)	3359 (67.31)	<0.001
	**2**	50,812 (19.81)	421 (30.51)	952 (19.08)	
	**3**	13,614 (5.31)	166 (12.03)	289 (5.79)	
	**4**	18,785 (7.32)	322 (23.33)	390 (7.82)	
	**Total**	256,502	1380	4990	
**Histology Subtype**	**Melanoma NOS**	192,595 (50.23)	1355 (56.88)	4148 (52.47)	<0.001
	**Acral Lentiginous**	4395 (1.15)	465 (19.52)	284 (3.59)	
	**Nodular**	35,842 (9.35)	186 (7.81)	640 (8.1)	
	**Superficial Spreading**	114,225 (29.79)	222 (9.32)	2124 (26.87)	
	**Lentigo Maligna**	19,685 (5.13)	24 (1.01)	390 (4.93)	
	**Others**	16,720 (4.36)	130 (5.46)	320 (4.05)	
	**Total**	383,462	2382	7906	

**Table 4 cancers-13-01455-t004:** Baseline characteristics of patients with ocular and mucosal melanoma (NCDB 2004–2015). H&N: head and neck; GU: genitourinary; GIT: gastrointestinal tract excluding anorectum.

Noncutaneous Invasive Melanoma (All Cases)	Ocular *n* = 17,713 (%)	Head and Neck *n* = 2589 (%)	Genitourinary *n* = 3255 (%)	Anorectal *n* = 1351 (%)	Gastrointestinal Tract, Excluding Anorectum *n* = 240 (%)	Chi-Squared *p*-Value
**Females**	8491 (47.94)	1299 (50.17)	3056 (93.89)	795 (58.85)	87 (36.25)	<0.001
**Age**						<0.001
≤30 Years	423 (2.39)	28 (1.08)	86 (2.64)	8 (0.59)	4 (1.67)	
31 to 64 Years	9153 (51.67)	899 (34.72)	1167 (35.85)	538 (39.82)	93 (38.75)	
≥65 Years	8137 (45.94)	1662 (64.19)	2002 (61.51)	805 (59.59)	143 (59.58)	
**Race**						<0.001
Caucasian	16,861 (95.19)	2307 (89.11)	2966 (91.12)	1191 (88.16)	217 (90.42)	
AA	178 (1.00)	149 (5.76)	159 (4.88)	61 (4.52)	11 (4.58)	
**Positive Lymph Nodes identified**	18 (0.11)	259 (10.38)	597 (18.69)	299 (22.84)	56 (24.35)	<0.001
**Metastasis present at diagnosis**	326 (1.90)	338 (13.88)	230 (7.48)	233 (19.56)	72 (33.03)	<0.001
**Died**	5488 (30.98)	1806 (69.76)	1969 (60.49)	1003 (74.24)	169 (70.42)	<0.001
**Received Radiation Therapy**	12,256 (69.33)	1423 (55.37)	570 (17.64)	323 (24.14)	34 (14.35)	<0.001
**Underwent Surgery**	5852 (33.08)	2149 (83.10)	2849 (87.58)	1061 (78.77)	158 (65.83)	<0.001
**Received Systemic therapy**	612 (3.47)	520 (20.40)	576 (17.88)	460 (34.51)	72 (30.13)	<0.001

**Table 5 cancers-13-01455-t005:** Survival analysis and comparison of median OS (overall survival) of melanoma cases with metastatic disease at presentation (by subtype).

Survival Analysis	Cutaneous	Ocular	Mucosal
Median OS 2004–2010	8.11 months	9.36 months	7.36 months
Median OS 2011–2015	10.35 months	9.10 months	10.71 months
3-year survival 2004–2010	14.64%	11.36%	5.53%
3-year survival 2011–2015	21.10%	11.88%	8.60%
**Comparison of Median OS**	**Unadjusted HR**	**CI**	***p*-value**
Cutaneous: 2011–2015 vs. 2004–2010	0.82	0.79–0.85	<0.001
Ocular: 2011–2015 vs. 2004–2010	1.03	0.81–1.31	0.797
Mucosal: 2011–2015 vs. 2004–2010	0.74	0.65–0.86	<0.001
Mucosal vs. Ocular	1.03	0.90–1.18	0.645
Ocular vs. Cutaneous	1.14	1.01–1.28	0.029
Mucosal vs. Cutaneous	1.17	1.09–1.26	<0.000

**Table 6 cancers-13-01455-t006:** Multivariate logistic regression analysis for sociodemographic factors associated with the presence of metastasis at diagnosis for cases of invasive melanoma.

Factors	OR	95% CI	*p*-Value
**Age**	1.01	1.00–1.01	<0.001
**Female vs. Male**	0.72	0.7–0.74	<0.001
**Race**			
African Americans vs. Caucasians	2.37	2.11–2.67	<0.001
Others vs. Caucasians	1.06	0.95–1.17	0.313
**Invasive melanoma subtype**			
Ocular vs. Cutaneous	0.37	0.33–0.41	<0.001
Mucosal vs. Cutaneous	2.46	2.28–2.66	<0.001
**Charlson Deyo score**			
1 vs. 0	1.43	1.37–1.49	<0.001
2 vs. 0	2.05	1.91–2.2	<0.001
3 vs. 0	3.48	3.16–3.83	<0.001
**Socioeconomic Status: *Median household income for patient’s zip code***			
40,227–50,353 (level 2) vs. <40,227 (level 1)	1.03	0.96–1.1	0.372
50,354–63,332 (level 3) vs. <40,227 (level 1)	0.99	0.93–1.06	0.785
>46,000 to >63,333 (level 4) vs. <40,227 (level 1)	0.97	0.91–1.05	0.476
**Number of adults in the patient’s zip code who did not graduate from high school**			
20–28.9% vs. >29%	0.91	0.85–0.96	<0.001
14–18.9% vs. >29%	0.83	0.78–0.89	<0.001
<14% vs. >29%	0.75	0.71–0.81	<0.001
**Treating Facility Type**			
Academic/research program vs. Community cancer program	0.76	0.74–0.79	<0.001
Integrated network cancer program vs. Community cancer program	1.01	0.97–1.06	0.65
**Insurance Status**			
Private insurance vs. Not insured	0.35	0.32–0.37	<0.001
Medicaid vs. Not insured	1.26	1.15–1.39	<0.001
Medicare or other government insurance vs. Not insured	0.44	0.4–0.47	<0.001
**Geographical location**			
West vs. North East	1.09	1.04–1.15	<0.001
Midwest vs. North East	1.01	0.97–1.06	0.53
South vs. North East	1.06	1.01–1.1	0.01

## Data Availability

The primary dataset (National Cancer Database) is available publicly through the American College of Surgeons (https://www.facs.org/quality-programs/cancer/ncdb). The datasets generated and/or analyzed during the current study are available from the corresponding author on reasonable request.

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
