# Peer review of "NCDB Analysis of Melanoma 2004–2015: Epidemiology and Outcomes by Subtype, Sociodemographic Factors Impacting Clinical Presentation, and Real-World Survival Benefit of Immunotherapy Approval"

_cancers, 2021, doi:10.3390/cancers13061455_

Round 1
Reviewer 1 Report
1.Overall survival was the primary outcome, but for the research questions, disease-specific survival would be of interest. Possibly cause of death was not available. Could the authors calculate the relative survival? Regression analyses on relative survival are possible as well and the relative excess risk of dying can be calculated instead of the hazard ratio.
2.In table 1 is reported how many patients had positive lymph nodes. Is there also information about how many patients underwent an SLN biopsy?
3.There is no information about Breslow thickness or ulceration. Could the authors explain in the methods, why this was not available? It should also be mentioned in the limitations, as these are important prognostic factors and a lack of information could easily lead to residual confounding. For example, melanomas of African Americans may be thicker, which might explain the poor prognosis.
4.The authors present the number of cases in figure 2. What was the reason to present absolute numbers? Because, age standardized incidence rates would provide information about the increase of incidence, regardless of an aging population.
Reviewer 2 Report
This is an important study that aims to evaluate the differences in clinical outcomes pre/post the immunotherapy era, with attention to the impact of sociodemographic factors on survival outcomes. -Overall, the study of socioeconomic factors and demographics on the impact of cancer care is a very important topic and I applaud the authors for investigating this. However, the data is not always clearly presented, and it makes it challenging at times to understand exactly which groups of patients were compared. The title of the paper implies that epidemiology and the socioeconomic/demographic factors are the primary objective of the paper. The background also states that they are planning to describe the impact of the approval of immune checkpoint inhibitors on survival based on melanoma subtype and race. However, the primary outcome listed in the methods is overall survival of metastatic melanoma calculated for each subtype. It appears that the paper is largely directed toward the differences in the melanoma subtypes (cutaneous/ocular/mucosal) – in which case the paper should be titled differently.
Several additional questions arise while reading the manuscript:
- Regarding facility type, it would be interesting to know if there is a socio/demographic difference in patients who have access to academic institutions vs community. For example, academic facility is associated with better survival per their findings; however, this is not further explored based on demographics or socioeconomic factors.
- Table S2, the total number of Caucasian and African American patients should be included at the top. They should also assess for statistical significance between the relative proportions of histology subtype.
- Table S3 should have the total number of lymph nodes positive at presentation and metastatic disease at presentation. This table should also be further broken down by race. Additionally, it is not clear if the lymph node positive and metastatic disease patients are mutually exclusive or if some of the lymph node positive patients are also captured in the metastatic disease category?
- In table S3, who does the died column refer to? Because for example if you add the melanoma OS row together for lymph node positive and metastatic disease together you get approx 41k, but the died column says 65K. I’m not sure which additional patients the died column is capturing, but the study question wants to assess the impact of ICI on mortality, then survival data should only be for metastatic patients as in this time frame adjuvant ICI was only in a clinical trial setting.
- The survival improvement noted in Caucasians was statistically significant, the survival difference in AA patients was a similar value, but not statistically significant – however, in the conclusions the authors state that there is a survival improvement in Caucasians without one in AA pts, and this conclusion seems to overstate the data presented. It is possible, given the low number of patients the authors are unable to find statistical significance in the AA patients – however this is never explored as a possibility. Furthermore, in Table S6 (adjusted HR for survival with metastatic melanoma post ipi v prior to ipi, in the race category comparing Caucasian and African American patients the adjusted HR is 1.08 and is not statistically significant).
- On page 8, the section for impact of race on survival states that Kaplan Meier curves shown in figure 3, however, there is no Kaplan Meier assessment done by race.
The introduction mentions data collected from 2005 to 2014, but elsewhere (such as the Methods) it states 2004-2014, and then elsewhere states comparison is made 2004-2011 vs. 2011 vs. 2015.
Which AJCC edition was used for staging?
75.91% had invasive melanoma. By definition, melanoma is an invasive process. Did the remaining 24% have melanoma in situ (stage 0 melanoma), which is in fact a pre-malignant and not actually malignant process?
Almost 54% of the cutaneous melanomas fall under ‘NOS’ or ‘other’. How is this accounted for in the analysis, considering there may be mucosal or other subtypes there?
Why isn’t the breakdown for cutaneous included in Figure 1, as it is for non-cutaneous?
‘The median overall survival for all invasive cases of cutaneous melanoma was 161.12 months (approximately 13.5 years), 122.91 months (about 10 years) for ocular melanoma, and 26.29 months (approximately 2 years) for mucosal melanoma.’ Is this for all cases from 2004-2015? What about by stage?
Most patients with stage IV disease had earlier stage disease previously, with metastatic recurrence. How is this accounted for?
‘The low frequency of lymph node positivity in the ocular melanoma cohort can be explained by absence of lymphatic supply in the uvea’. It is important to note also that sentinel lymph node biopsy is not standard of care for ocular melanoma, and not typically checked.
‘This observation could possibly be explained by referral bias and higher index of suspicion, coupled with better logistics in terms of diagnostic capabilities at academic/research institutions.’ Academic centers may also be more likely to longitudinally follow earlier stage patients, and thus catch recurrences earlier.
Metastatic melanoma of unknown primary occurs in up to 10% of metastatic patients. In that case, it is almost impossible to histologically differentiate between cutaneous and mucosal. Is this accounted for? If not, it should be discussed or referred to.
As the target of ipilimumab is mentioned in the intro (CTLA-4), the target of pembrolizumab and nivolumab should be mentioned also.
while it may be beyond the scope of this paper, the authors are not analyzing the difference in outcomes in the post-PD1 era (pembro/nivo/ipi+nivo), all approved in 2015 or after. These are clinically superior drugs and one might argue this is the more interesting research question.
Minor:
‘For example, amongst anorectal melanoma patients, systemic therapy was a part of treatment in 34.51% and 17.32% of cases, respectively.’ It is not clear what two groups are being compared here.
‘While 41.61% of the cutaneous melanoma cases were treated in community centers. The majority cases of ocular (77.49%) and mucosal melanoma (51.94%) patients received treatment at an academic/research in- stitution, and we postulate that this reflects the unique, highly specialized care needs of these patients.’ There should be a comma, not a period separating these two statements (between ‘center’s and ‘the majority’.
‘treatment at academic facility’- missing ‘an’
Round 2
Reviewer 2 Report
My concerns have been adequately addressed. For several points, they say 'we will add' or 'will change' this in the discussion- assuming this is done, I think this is acceptable for publication.